# A Thermodynamic Constitutive Model for Saturated Sand

**DOI:** 10.3390/e21020136

**Published:** 2019-02-01

**Authors:** Shize Xiao, Xiaohui Cheng, Zhou Yang

**Affiliations:** Department of Civil Engineering, Tsinghua University, Beijing 100084, China

**Keywords:** non-equilibrium thermodynamics, sand, constitutive model, triaxial undrained tests

## Abstract

This paper establishes a non-equilibrium thermodynamic constitutive model that can predict the undrained shear behavior of saturated sand. Starting from the basic laws of thermodynamics, the model does not require the classical concepts in elasto-plastic models, such as the yield function, the flow rule, and the hardening rule. It is also different from the existing thermodynamic constitutive models in soil mechanics literatures. The model does not use a complex nonlinear elastic potential as usually and introduces a coupling energy dissipative mechanism between the viscosity and elasticity relaxation, which is essential in granular materials. Then this model was used to simulate the undrained shear test of Toyoura sand. The model can predict the critical state, dilatancy-contraction and hardening-softening characteristics of sand during undrained triaxial shearing.

## 1. Introduction

In the deformation process, granular materials show not only the characteristics of elastic solids, such as a certain degree of shear stiffness or the ability to propagating shear waves, but also the rheological properties of viscous fluids, such as the liquefaction and steady shear flow [1]. Physical and geotechnical studies of granular materials have shown that the solid-liquid transition mechanism, that is, the phase transition law, of granular materials can been affected by (1) particle packing fraction, (2) mean effective stress level, (3) frequency of external excitation, and (4) pore fluid-particle interactions. The solid-liquidlike rheological behavior presents more complex and varied properties when the granular materials are near the phase transition point at very low or zero effective stress [2,3]. For example, the shear thickening phenomenon in granular physics research, and the complete liquefaction of sand in soil mechanics. Therefore, the research on the rheological behavior of granular material in the low effective stress region has received extensive attentions from physics and engineering researchers. It is not only one of the fundamental problems in the physical study of granular materials, but also the important analysis basis of liquefaction and post-liquefaction deformation of the lifeline infrastructures.

Researchers have carried out several experimental studies under various stress conditions of sands, which shows the granular material exhibits two unusual physical properties near the phase transition point. One is the so called significantly increased apparent friction angle problem. Sture et al. (1998) [2] performed triaxial tests on the dry Ottawa sand in the NASA Space Shuttle under conditions of microgravity (low effective stress). The results show that the granular material exhibits extremely high friction angles and dilatancy angles. They speculate that it may be related to the inter-locking mechanism of sand particles under low confining pressure. Towhata et al. (1999, 2005, and 2010) [3,4,5] performed three experiments of Toyoura sand, the embedded rod dragging experiment in the shaken liquefied Toyoura sand under 1g conditions, the triaxial shear rheological experiment with low effective stress under 1G conditions, and the triaxial shear rheological test in the falling tower which can provide a microgravity environment. The results show that the granular material exhibits some of Bingham fluid properties in the low effective stress region. However, the apparent viscosity coefficient exhibits a highly nonlinear relationship with the shear rate. It is completely different from a classical Bingham fluid.

The solid-liquid-like nature of the granular material near the “zero effective stress” state is similar to the elusive nature of liquefaction and large post-liquefaction deformation called the “flowing sand” in soil mechanics and geotechnical engineering. When saturated sand is under earthquake or other undrained shear process, the effective stress decreases with the pore water pressure rises, the sand will show more liquid-like mechanical properties. “Flowing sand” can cause serious damage to lifeline infrastructures embedded in it, such as power and water transmission lines, tunnels, and bridge piles. Therefore, in the study of geotechnical earthquake engineering, the research about liquefaction of saturated sand and post-liquefaction deformation has been conducted for more than half a century. It has long been the core research content of soil mechanics and geotechnical earthquake engineering. In Japan, even based on the above studies, the Japanese Seismic Design Specifications for Highway Bridges (1966) regulates in simple rule of thumb the horizontal thrust of the pile foundation caused by the “flowing sand” after liquefaction. To establish a continuous constitutive model with better prediction for sand liquefaction is the consistent voice of the scientific and engineering community for many years.

This paper abandons the classical solid mechanics or soil mechanics approach, and does not follow the thermodynamic method proposed by Houlsby and Puzrin (2000) in geo-mechanics [6,7], but the theory of granular solid hydrodynamics proposed in granular physics [8]. This newly-developed model, based on the non-equilibrium thermodynamics theory, attempts to describe the quasi-static critical state or generalized rheology of sands.

In the previous research, the Tsinghua research group has established a thermo-hydro-mechanical coupling thermodynamic constitutive model (referred to as Tsinghua Thermodynamics Soil Model, TTS) for clays [9,10,11], which can describe nonlinear deformation, failure, creep, stiffness degradation, cyclic loading response, stress induced anisotropy, and temperature coupling effects of clays. However, the uniform hydrodynamic constitutive model of sands has not yet been established.

The non-equilibrium thermodynamics takes the local equilibrium as the core assumption, and regards the thermodynamic state of an isolated system as the statistical sum of the micro-elements in thermal equilibrium. The thermodynamic state of elements can be completely and uniquely described by a set of independent state variables, and the entropy is introduced to describe the thermal motion intensity of the system. The system always evolves in the direction of “entropy increasing”, and the dynamic evolution of the system can be treated by extreme value analysis of entropy.

As a kind of complex material, granular material has a unique mesoscale compared with conventional materials (Newtonian fluid or crystalline solid). Macroscopic continuous theory such as classical fluid dynamics is no longer applicable at this scale. Starting from the framework of non-equilibrium thermodynamics, Jiang and Liu [12] introduced the granular entropy describing the mesoscale thermodynamic behavior of granular materials to describe the intensity of collision, slip, rolling and other particle motions at mesoscale. The TTS model further extends this concept to complex behaviors of geotechnical materials.

This paper first introduces the non-equilibrium thermodynamic dissipation mechanisms and constitutive equations derived for sands accounting for the coupling mitigation coefficient between the viscosity and the elastic relaxation in the thermodynamic dissipation mechanism. Based on this modified TTS model, the critical state (quasi static rheological state), and the dilatancy-contraction and hardening-softening properties before the critical state were qualitatively analyzed. Then, the undrained triaxial experiments of saturated Toyoura sand under medium and high consolidation pressure with different void ratios are simulated [1,13]. This paper demonstrated that the modified TTS constitutive model has good simulation ability for the key features of undrained sand, such as the critical state, the shear dilatancy-contraction and hardening-softening. In the future work, the model’s ability to simulate the rheological properties near the sand phase transition point will be further extended.

## 2. Modified TTS Model

### 2.1. Onsager Dissipative Force-Flux Relations

Non-equilibrium thermodynamic theory describes the mechanical characteristics of granular materials compared to general solid and liquid materials by selecting suitable independent state variables and specifying the dissipative mechanisms. These new theoretical features are summarized as so-called transient hyper-elasticity and granular fluctuation in granular materials.

The transient elasticity of the sand indicates that once the particle is deflected by the external disturbance, it deviates from the thermodynamic equilibrium state, generates inelastic deformation and is accompanied by an energy dissipating process. The particles will reach a new equilibrium state after the disturbance is over. That is, the granular material does not have a pure elastic stage as described in the classical elasto-plastic theory, but elastic energy is continuously dissipated during the disturbance process. This process is called elastic relaxation process resulting from the transient elastic properties of granular materials.

The theory of gas molecular dynamics uses the concept of entropy to quantitatively describe the energy dissipation in the system caused by the molecular irregular movement under the macroscopic incentives. Granular materials also have similar irregular motion mechanisms at the mesoscopic level (particle level). Under the macroscopic incentives, in addition to the macroscopic average kinetic energy, there will be a velocity shift between the particles inside the material, and this will cause interactions, such as collision, friction, vibration, and rolling between the particles, resulting in an additional dissipation mechanism. This phenomenon is called granular fluctuation. Jiang and Liu (2009) introduced the concept of granular entropy in the GSH theory, analogous to the entropy theory in the classical thermodynamic theory. The granular entropy increasing equation is used to quantitatively describe the dissipative process produced by granular fluctuation, as follows:
(1)$$\rho_{d}d_{t}s_{g}=R_{g}/T_{g}-I_{g}$$


The granular entropy increasing equation indicates that the increase in granular entropy will be determined by the granular entropy production rate $R_{g}/T_{g}$ and the conversion rate $I_{g}$ from granular entropy to real entropy. $R_{g}$ is the energy stimulated by the external excitation in the granular fluctuation. After selecting the appropriate dissipative force-flux relationship, it can be quantitatively described. Under isothermal conditions, the strain rate $d_{t}\varepsilon_{ij}$, the elastic stress $\pi_{ij}$, and the granular temperature $T_{g}$ are selected as dissipative forces in the original TTS model. The conjugated dissipative fluxes are viscous stress $\sigma_{ij}^{v}$, granular entropy stress $\sigma_{ij}^{g}$, elastic relaxation rate $Y_{ij}$, and entropy conversion rate $I_{g}$.

According to Onsager relation, a linear relationship between dissipative force and flux is satisfied near the equilibrium of thermodynamics, and then the relationship between them can be described by the mitigation coefficient matrix as follows:
(2)$$\left\{\begin{array}{c} \sigma_{ij}^{v}\\ Y_{ij}\\ I_{g}\\ \sigma_{ij}^{g} \end{array}\right\}=\left[\begin{array}{cccc} \eta_{ijkl}^{v} & -\alpha_{ijkl} & 0 & 0\\ \alpha_{klij} & \lambda_{ijkl} & 0 & 0\\ 0 & 0 & \gamma & 0\\ 0 & 0 & 0 & \eta_{ijkl}^{g} \end{array}\right]\left\{\begin{array}{c} d_{t}\varepsilon_{kl}\\ \pi_{kl}\\ T_{g}\\ d_{t}\varepsilon_{kl} \end{array}\right\}$$


In this coefficient matrix, the meaning of the four diagonal terms is: $\eta_{ijkl}^{v}$ is the viscosity coefficient, which represents viscous dissipation. Due to the small strain rate in the triaxial test, this coefficient can be ignored. $\lambda_{ijkl}$ is the elastic relaxation coefficient, which characterizes the irreversible effect of the particle system due to the elastic relaxation; γ is the granular entropy mitigation coefficient and characterizes the kinetic energy fluctuation due to the granular fluctuation. That is, the process of energy dissipation caused by particles’ random motions, such as friction and collisions. $\eta_{ijkl}^{g}$ is the mitigation coefficient corresponding to the strain rate of the particle skeleton, and it represents the elastic potential energy dissipation due to the process of particle fluctuation:
(3)$$\eta_{ijkl}^{v}=\eta_{s}^{v}\delta_{ik}\delta_{jl}+\left(\eta_{v}^{v}-\eta_{s}^{v}/3\right)\delta_{ij}\delta_{kl}$$
(4)$$\gamma =\gamma_{0}+\gamma_{1}T_{g}$$
(5)$$\eta_{ijkl}^{g}=\eta_{s}^{g}\delta_{ik}\delta_{jl}+\left(\eta_{v}^{g}-\eta_{s}^{g}/3\right)\delta_{ij}\delta_{kl}$$


The mitigation coefficient matrix also contains an off-diagonal element. Mathematically, the Onsager dissipative relations represent a Taylor series expansion of the dissipation function near the equilibrium state, and the off-diagonal elements of the matrix represent the coupling between two dissipative mechanisms. In this paper, the coupling coefficient $\alpha_{ijkl}$ between the viscosity and the elastic relaxation is introduced into the original TTS model. It is a new dissipative mechanism in addition to the viscosity, elastic relaxation, and particle fluctuation in granular materials. This dissipative mechanism affects macroscopic properties such as the critical state, the dilatancy-contraction, and the shear hardening-softening. The coefficient $\alpha_{ijkl}$ is suggested to adopt the following form, which contain the elastic deviatoric strain:
(6)$$\alpha_{ijkl}=\alpha_{0}\delta_{ik}\delta_{jl}-\alpha_{1}\left(e_{ij}^{e}\delta_{kl}+e_{kl}^{e}\delta_{ij}\right)$$


In the above expression, the coefficients $\alpha_{0}$ and $\alpha_{1}$ are parameters related to the sand properties. And $e_{ij}^{e}$ is the elastic deviatoric strain ($e_{ij}^{e}=\varepsilon_{ij}^{e}-\varepsilon_{kk}^{e}\delta_{ij}/3$). There are two major terms contributed to the generation of non-elastic strain, coupling dissipation and elastic relaxation, and the expression of elastic strain rate is also obtained:
(7)$$Y_{ij}=\alpha_{ijkl}d_{t}\varepsilon_{kl}+\lambda_{ijkl}\pi_{kl}$$
(8)$$d_{t}\varepsilon_{ij}^{e}=\left(1-\alpha_{0}\right)d_{t}\varepsilon_{ij}+\alpha_{1}e_{ij}^{e}d_{t}\varepsilon_{v}+\alpha_{1}e_{kl}^{e}d_{t}e_{kl}\delta_{ij}-\lambda_{s}T_{g}e_{ij}^{e}-\lambda_{v}T_{g}\varepsilon_{v}^{e}\delta_{ij}$$


### 2.2. Effective Stress Equation

Based on the theory of non-equilibrium thermodynamics of saturated soil, Zhang and Cheng (2015) proved that the total stress of the soil is:
(9)$$\sigma_{ij}=\pi_{ij}+\sigma_{ij}^{v}+\sigma_{ij}^{g}+p_{1}\delta_{ij}$$


In the formula, the elastic stress $\pi_{ij}$ can be obtained from the elastic potential energy function, and the viscous stress $\sigma_{ij}^{v}$ and the granular fluctuation stress $\sigma_{ij}^{g}$ can be obtained by the Onsager dissipation relation, and $p_{1}$ is the pore water pressure. According to the Terzaghi effective stress principle, Equation (9) can be further rewritten as an effective stress equation:
(10)$$\sigma_{ij}^{\prime }=\left(1-\alpha_{0}\right)\pi_{ij}+\alpha_{1}e_{ij}^{e}\pi_{kk}+\alpha_{1}e_{kl}^{e}\pi_{kl}\delta_{ij}+\left(\eta_{ijkl}^{v}+\eta_{ijkl}^{g}\right)d_{t}\varepsilon_{ij}$$


In the formula, $\sigma_{ij}^{\prime }$ is the effective stress. Due to the low strain rate of sand in triaxial shear tests, the effective stress term associated with the strain rate, which is $(\eta_{ijkl}^{v}$+$\eta_{ijkl}^{g})d_{t}\varepsilon_{ij}$, can be omitted.

### 2.3. Constitutive Equations under Undrained Triaxial Condition

Under undrained triaxial conditions, there is no volumetric strain, the axial strain is controlled by the loading conditions, $\varepsilon_{2}=\varepsilon_{3}=-\varepsilon_{1}/2$, and the remaining elements of strain tensor is zero. When ignoring the compressibility of water and sand, the density will remain constant and the mass conservation equation can be ignored for undrained triaxial tests.

(1) Equation of granular entropy increasing

In the formula, $\rho_{d}$ is the dry density of sand, $e_{ij}=\varepsilon_{ij}-\varepsilon_{v}\delta_{ij}/3$ is deviatoric strain, $\gamma =\gamma_{0}+\gamma_{1}T_{g}$ is the granular mitigation coefficient:
(11)$$b\rho d_{t}T_{g}=\eta_{s}^{g}d_{t}e_{ij}d_{t}e_{ij}-\gamma_{0}T_{g}-\gamma_{1}T_{g}^{2}$$


(2) Equation of elastic relaxation

Among the undrained triaxial shearing process, $\varepsilon_{ij}^{e}=0\left(i\neq j\right)$, the elastic strain tensor, therefore, only has two independent variables, which are $\varepsilon_{11}^{e}$ and $\varepsilon_{33}^{e}$. Then the elastic relaxation equation can be simplified as:
(12)$$d_{t}\left(\varepsilon_{11}^{e}-\varepsilon_{33}^{e}\right)=1.5\left(1-\alpha_{0}\right)d_{t}\varepsilon_{11}-\lambda_{s}T_{g}\left(\varepsilon_{11}^{e}-\varepsilon_{33}^{e}\right)$$
(13)$$d_{t}\varepsilon_{v}^{e}=3\alpha_{1}\left(\varepsilon_{11}^{e}-\varepsilon_{33}^{e}\right)d_{t}\varepsilon_{11}-3\lambda_{v}T_{g}\varepsilon_{v}^{e}$$


(3) Elastic potential energy density function

The elastic potential energy density function can be defined by the four functional forms of Internal energy, Helmholtz free energy, enthalpy, and Gibbs free energy. These definitions can be transformed by Legendre transformation. This paper uses the Helmholtz free energy function to define the elastic potential energy function. In the original TTS model, the nonlinear elastic potential function proposed was selected. However, in this paper, the simplest linear elastic model is adopted:
(14)$$\omega_{e}=\frac{1}{2}K{\left(\varepsilon_{v}^{e}\right)}^{2}+G{\left(\varepsilon_{s}^{e}\right)}^{2}$$


(4) Equation of effective stress

Based on the elastic relaxation equation and elastic potential energy density function, the effective stress equation under undrained triaxial shear conditions can be obtained as follows:
(15)$$p^{\prime }=\left(1-\alpha_{0}\right)K\varepsilon_{v}^{e}+\frac{4}{3}\alpha_{1}G{\left(\varepsilon_{11}^{e}-\varepsilon_{33}^{e}\right)}^{2}$$
(16)$$q=\left[3\alpha_{1}K\varepsilon_{v}^{e}+2G\left(1-\alpha_{0}\right)\right]\left(\varepsilon_{11}^{e}-\varepsilon_{33}^{e}\right)$$


From the effective stress equations, influence of the coupling mitigation coefficient is manifested by the compression-shear coupling effect, and both the elastic shear strain $\varepsilon_{11}^{e}-\varepsilon_{33}^{e}$ and the elastic volumetric strain $\varepsilon_{v}^{e}$ will affect the mean effective stress and shear stress. At the same time, in the elastic strain equation, the elastic volumetric strain rate is also affected by the elastic shear strain. Without considering the coupling mitigation coefficient, the development of the elastic strain is only affected by the elastic relaxation mechanism, and the compression-shear coupling effect of the material cannot be considered in the effective stress equation, resulting in that the dilatancy-contraction and hardening-softening characteristics of the material cannot be reflected.

### 2.4. Critical States of Sand

The critical state is the quasi-static steady state that the soil material reaches after large deformation [14,15,16,17]. It shows that under the condition of constant shear stress, the material will continue to shear under constant shear rate, and keep the volume and effective stress unchanged. The critical state of sand is not affected by the initial consolidation condition of the samples, and shows rate-independence.

Since the effective stress of the material remains unchanged in the critical state, the elastic strain rate $d_{t}\varepsilon_{ij}^{e}=0$ and the granular entropy change rate $d_{t}T_{g}=0$. For the sand material with very slow shear rate under undrained triaxial shearing, the parameters of the critical state show rate-independence, and the entropy mitigation coefficient $\gamma_{0}$ is neglected in this case. From the constitutive equations, the analytical solution of the sand’s critical state can be obtained:
(17)$${\left(T_{g}\right)}_{cr}=\sqrt{\frac{\eta_{s}^{g}}{\gamma_{1}}}d_{t}\varepsilon_{s}$$
(18)$${\left(\varepsilon_{11}^{e}-\varepsilon_{33}^{e}\right)}_{cr}=\frac{\sqrt{1.5}m\left(1-\alpha_{0}\right)}{\lambda_{s}}$$
(19)$${\left(\varepsilon_{v}^{e}\right)}_{cr}=\frac{m^{2}\left(1-\alpha_{0}\right)\alpha_{1}}{\lambda_{s}\lambda_{v}}$$
(20)$${p^{\prime }}_{cr}=\frac{\left(1-\alpha_{0}\right)m^{2}K}{\lambda_{s}\lambda_{v}}\alpha_{1}+\frac{2{\left(1-\alpha_{0}\right)}^{2}m^{2}G}{\lambda_{s}^{2}}\alpha_{1}$$
(21)$$q_{cr}=\frac{\sqrt{13.5}{\left(1-\alpha_{0}\right)}^{2}m^{3}K}{\lambda_{s}^{2}\lambda_{v}}\alpha_{1}^{2}+\frac{\sqrt{6}{\left(1-\alpha_{0}\right)}^{2}mG}{\lambda_{s}}$$


In the above formula, $m={\left(\eta_{s}^{g}/\gamma_{1}\right)}^{0.5}$ as a constant.The stress ratio of ${p^{\prime }}_{cr}/q_{cr}$ will be a constant as the mean effective stress and deviator stress can both be expressed by a migration factors. The coupling mitigation coefficient is very important for the critical state. In the critical state, the material undergoes a pure shear deform condition. If this coefficient is not considered, the elastic volumetric strain will gradually decrease with the elastic relaxation process, which means that the effective mean stress will also decrease to zero. Therefore, the coupling coefficient is one of the necessary items to describe the quasi-static rheological state of the granular materials.

### 2.5. Loading Process

In the undrained triaxial shear test of sand, there are two issues as concerned as the dilatancy-contraction, hardening-softening characteristics. As shown in the above formula, the coupling mitigation coefficient introduced in this paper will produce a compression-shear coupling effect in the elastic relaxation equation and effective stress equations. In the elastic relaxation equation, the development of the shear strain will have a coupling effect on the volumetric elastic strain, and since the critical state and the volumetric elastic strain level of the initial state are directly determined by the material density and the initial consolidation conditions, the compression-shear coupling effects may make the elastic volumetric strain show a non-monotonic process during the development from the initial state to the critical state.

That is, at the beginning of shearing, the volumetric elastic strain rate $d_{t}\varepsilon_{v}^{e}<0$ due to the small shear elastic strain. With the development of the shear elastic strain, the volumetric elastic strain will show an increasing trend later. On the other hand, the compression-shear coupling effect in the effective stress equation will also be affected by $\alpha_{ijkl}$. The non-monotonicity of the elastic strain development and the compression-shear coupling characteristics in the effective stress equation can directly lead to the complex characteristics, such as dilatancy-contraction and peak shear strength.

## 3. Model Predictions for Undrained Triaxial Test of Sand

Based on the modified TTS model, this paper simulates the undrained triaxial test of Toyoura sand. Sand samples were undrained sheared under strain control conditions ($d_{t}\varepsilon_{1}=1.66\times 10^{-4}s^{-1}$). Multiple sets of experiments with different void ratio and initial consolidate condition of samples were performed in parallel. As mentioned above, the material parameters will be affected by the material density, so the parameters involved in this simulation are shown in Table 1.

Eleven sets of undrained triaxial shear experiments of saturated Toyoura sand were simulated. The void ratio and initial consolidation pressure of sand samples varied within wide range (Figure 1, Figure 2 and Figure 3). Taking Figure 1 as an example, the dense sand with a void ratio of e = 0.735 was tested under conditions of initial consolidation pressures of 100 kPa, 1000 kPa, 2000 kPa, and 3000 kPa, and all the samples exhibited the same extreme conditions when reached large strain level. That is, the critical state of the sample is independent of the initial consolidation conditions.

Another set of experiments was performed on a medium dense sand sample with a void ratio of e = 0.833. In Figure 2, Initial consolidation state will significantly affect the development trend of the mean effective stress $p^{\prime }$ and deviator stress q during the development of the strain. When the sample is subjected to a higher initial consolidation pressure, such as 2 MPa and 3 MPa, the peak of the deviator stress q appears at the shear strain of 2–4%, and falls back to the level at critical state. In addition, the initial consolidation condition will also affect the development path of mean effective stress $p^{\prime }$. Taking the experiments with initial consolidation pressure of 1 MPa and 2 MPa as examples, when the initial consolidation pressure is small, the sample will go through a contractive phase and then change to a dilative one. On the contrary, it continues to contract. In combination with the stress development path of all the experiments.

The dilatancy-contraction and hardening-softening process of the samples will be affected by the void ratio and the initial consolidation condition, and the simulation results are also in good agreement with the above process. If based on linear elastic model, classical associated elasto-plastic models do not have the ability to simulate the above phenomena. This also further validates the importance and necessity of the coupling mitigation coefficient $\alpha_{ijkl}$ introduced in this paper.

### 3.1. Prediction of Critical States

In the all 11 sets of simulations, the samples have the same void ratio with different initial confining pressures will reach the same critical state under large strain conditions (20–25%). The sample reaches a quasi-static rheological state with constant volume, constant effective stress, and constant shear stress. As shown in Figure 4, at critical states, a constant friction angle between the mean stress and the shear stress in the simulation results is consistent with the experimental result.

### 3.2. Loading Process Analysis

As described in Section 3.1, after introducing the coupling coefficient $\alpha_{ijkl}$, the model canpredict the complex dilatant/contraction and hardening-softening processes of sand. For example, a dense sand sample with void ratio e = 0.735 and initial consolidation pressure ${p^{\prime }}_{0}=3000\;\mathrm{kPa}$, the sample first enters acontractive zone at the beginning of shearing, and then enters a dilatation zone when the strain reaches about 3%. In Figure 5, the granular entropy temperature and elastic shear strain continue to increase until reaches the constant value at critical state, and the elastic volumetric strain first decreases due to the elastic relaxation dissipation. At this time, the mean effective stress decreases, and shear contraction occurs. Then, due to the increase of the elastic shear strain, the elastic volumetric strain increases under the influence of the coefficient $\alpha_{1}$, and the material exhibits a dilatancy trend.

For over-consolidated medium dense sand sample with void ratio e = 0.833 and initial consolidation pressure ${p^{\prime }}_{0}=3000\;\mathrm{kPa}$, the shear stress peaked when the axial strain approximately reached 3%, and strain softening occurred afterwards. In Figure 6, the shear elastic strain and shear volumetric strain increase and decrease monotonically in the shearing process, but due to the elastic volumetric strain is still at a high level, the material exhibits strain hardening state.

Then the elastic volumetric strain continues to decrease, the elastic shear strain gradually enters a steady value, and the material will appear to be in a softening state and eventually stabilize in a critical state. Compression-shear coupling effect generalized by $\alpha_{1}$ in the elastic relaxation and effective stress equations has a complex effect on the deformation characteristics of the material during the shearing process, and the shear deformation trend is related to the density and the initial consolidated pressure.

## 4. Discussions

The constitutive model of geo-materials based on classical elastoplastic mechanics cannot form a unified theoretical framework. In order to solve different problems, different concepts and amendments need to be introduced. In addition, the elastoplastic model lacks rigorous theoretical basis and may be contrary to the second law of thermodynamics.

The classical constitutive model based on thermodynamic theory framework uses the thermodynamic principle to obtain the yield function and flow function of geo-materials, which can fully satisfy the first and second laws of thermodynamics, and avoid the theoretical defects of plastic potential function in classical elastoplastic model. Thermodynamic models usually define corresponding independent state variables for all dissipative mechanisms, which are used to construct the corresponding dissipative potential functions. This descriptive method of energy dissipation is actually an imitation of the elastic potential function, and has not strictly proved the selection of the dissipative mechanism.

The theoretical framework of hydrodynamics used in TTS model, is based on the laws of conservation and non-equilibrium thermodynamics. In this theoretical framework, there is a well-established quantitative theory for determining dissipative forces and dissipative flows, namely the Onsager migration coefficient model. In addition, the TTS model following GSH introduces particle entropy to describe the fluctuation motion of granular particles and the dissipation caused by it. This model considers the connection and difference between the macroscopic behavior and the mesoscopic behavior of the material in a unified constitutive functions. In this model, elastic strain and inelastic strain are always accompanied, without the concept of yielding surface. This is closer to the true mechanical characteristics of geo-materials.

## 5. Conclusions

The modified TTS model is a constitutive model that does not require yield surface, flow rule, loading/unloading criteria, and hardening-softening criteria. The quantitative description of energy dissipation is attributed to the determination of a series of Onsager mitigation coefficients.

The coupling mitigation coefficient $\alpha_{ijkl}$ introduced is the essential in modeling the critical state of sands. Through the prediction of undrained triaxial shear experiments on saturated Toyoura sand, the following conclusions are drawn:
(1)The model can predict the critical state characteristics of sand, including the rate-independence of critical state, the void ratio–mean effective stress relationship, and the frictional angle at critical state. When $\alpha_{ijkl}$ is not considered, the elastic volumetric strain will gradually decrease with the granular fluctuation when the material is at a critical state under pure shearing condition, and the critical state cannot be correctly described.(2)Without the definition of complex non-associate flow rule, the modified TTS model has the ability to simulating the dilatancy-contraction and hardening-softening phenomenon during undrained shearing of sand. This ability is mainly derived from the compression-shear coupling provided by the coupling mitigation coefficient $\alpha_{ijkl}$ introduced in this paper, which makes the model exhibit different dilatancy-shearing mechanism and softening-hardening mechanism under different mean effective stress levels.


## Figures and Tables

**Figure 1 entropy-21-00136-f001:**
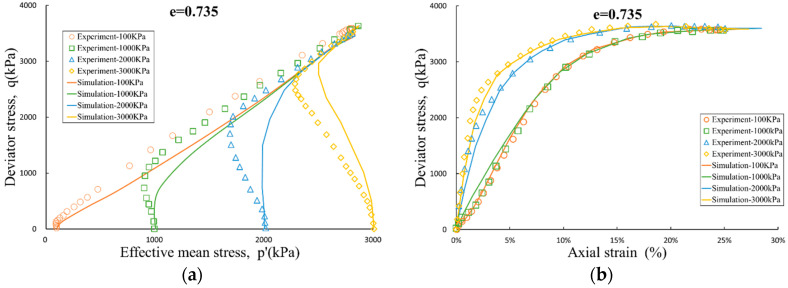
Undrained triaxial test and simulation for e = 0.735: (**a**) effective stress path; and (**b**) stress-strain curve.

**Figure 2 entropy-21-00136-f002:**
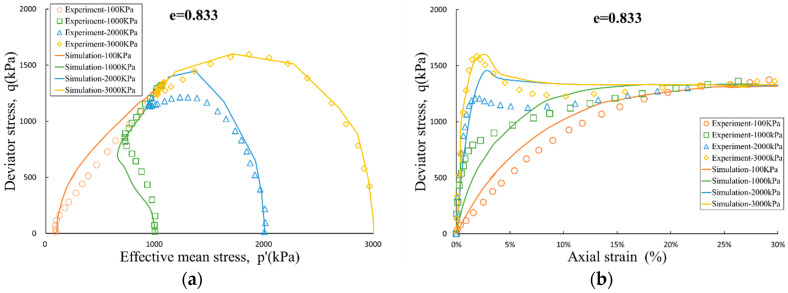
Undrained triaxial test and simulation for e = 0.833: (**a**) effective stress path; and (**b**) stress-strain curve.

**Figure 3 entropy-21-00136-f003:**
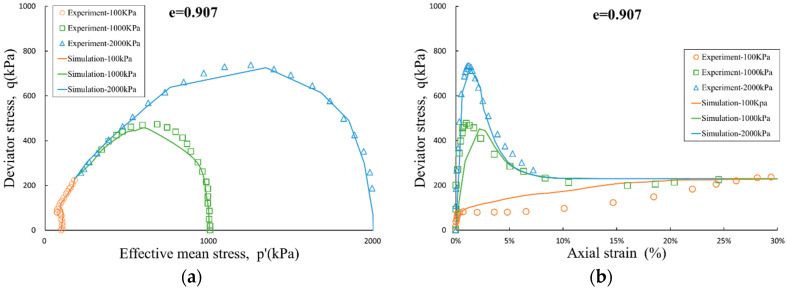
Undrained triaxial test and simulation for e = 0.907: (**a**) effective stress path; and (**b**) stress-strain curve.

**Figure 4 entropy-21-00136-f004:**
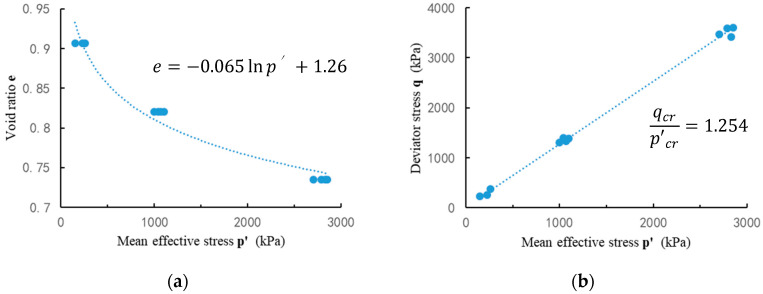
Simulation results of critical state of Toyoura sand: (**a**) relation between mean effective stress and void ratio; and (**b**) frictional angle at critical states.

**Figure 5 entropy-21-00136-f005:**
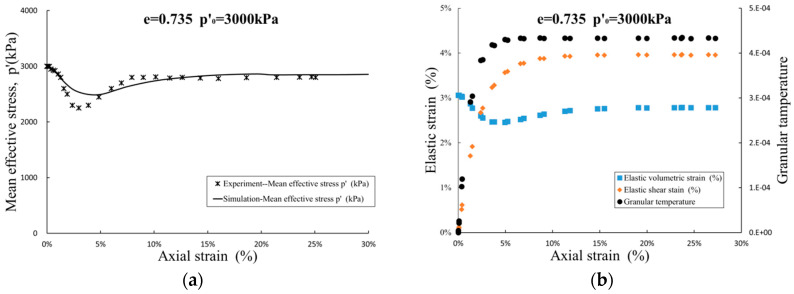
Result of simulation for sand with e = 0.735, ${p^{\prime }}_{0}=3000\;\mathrm{kPa}$: (**a**) mean effective stress-strain path; and (**b**) development of elastic strain and granular temperature.

**Figure 6 entropy-21-00136-f006:**
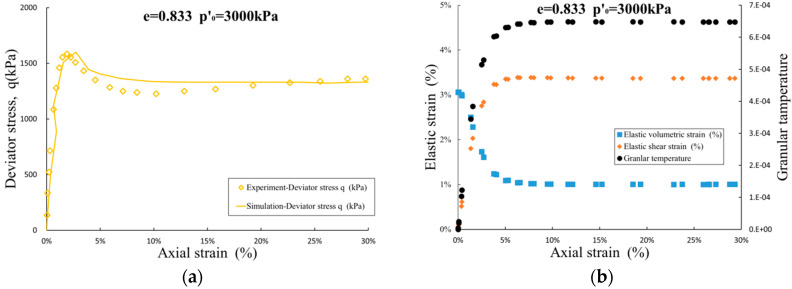
Result of simulation for sand with e = 0.833, ${p^{\prime }}_{0}=3000\mathrm{kPa}$: (**a**) deviator stress-strain path; and (**b**) development of elastic strain and granular temperature.

**Table 1 entropy-21-00136-t001:** The value of parameters used in each set of experiment prediction.

No.	e	$D_{r}$	${p^{\prime }}_{0}/kPa$	K/kPa	G/kPa	$\alpha_{0}$	$\alpha_{1}$	$\eta_{s}^{g}$	$\gamma_{1}$	$\lambda_{s}$	$\lambda_{v}$
1	0.735	32.9%	100	98,000	6000	0	6.09	199.64	8.65	11.54	5.97
2	0.735	32.9%	1000	98,000	6000	0	5.89	191.82	8.45	12.96	6.87
3	0.735	32.9%	2000	98,000	6000	0	9.24	198.38	9.52	13.15	4.92
4	0.735	32.9%	3000	98,000	6000	0	9.64	199.45	9.19	14.61	5.28
5	0.833	60.9%	100	98,000	6000	0	1.09	111.64	53.42	9.74	5.24
6	0.833	60.9%	1000	98,000	6000	0	5.82	174.06	59.10	15.54	12.81
7	0.833	60.9%	2000	98,000	6000	0	10.02	200.45	10.21	11.93	8.03
8	0.833	60.9%	3000	98,000	6000	0	9.29	178.14	11.18	11.47	8.02
9	0.907	82.0%	100	98,000	6000	0	5.3	232.6	16.7	21.0	10.5
10	0.907	82.0%	1000	98,000	6000	0	5.0	198.4	7.5	15.1	7.2
11	0.907	82.0%	2000	98,000	6000	0	7.9	197.2	7.1	16.2	10.4

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
