# Peer review of "A Thermodynamic Constitutive Model for Saturated Sand"

_entropy, 2019, doi:10.3390/e21020136_

Reviewer 1 Report

1 Introduction

Authors indicated some key futures as below;

Gradual materials

Soil particle packing friction

Fluid pore pressure

External frequency loading

Mean effective stress on effective stress theory for fully saturated soils

However, this study performance to NOT combine friction parameters of soils in discussions or consideration.

1 Introduction Page 1 8 line from bottom

Author said “lack of zero effective stress” in granular material properties. Reviewer NOT has agreement this sentence. In literatures, there are significant many papers related to zero effective stress, that is NOT enough to investigation of reference.

Page 1 from 2 line bottom

and post liquefaction. Please insert one space between “and” and “liquefaction”

This paper progressive using TTS established by Tsinghua research group. It is good. However, what is new portion or authors what improvement ?

Is correspond to models to clay applied to saturated sands?

This paper applied to Toyora sand NOLY.  For making stable of applying of suggested model, it should compare another gradual sand. 

In modified TTS model, dissipating process was improved. Other hands, untrained condition triaxial test was conducted out under untrained condition as no volume strain.  Are there dissipating ?

In modified TTS model, critical state is key feature to indicate accurate data sets. Please make clear in definition to critical state.

2.5. Loading process

Below sentence,

the critical state and the volumetric elastic strain level of the initial state are directly determined by the material density and the initial consolidation conditions,

Untrained triaxial compression test could not occurred strain. Is it necessary to improve elastic strain during compression or progressive of mean effective stress ?

Page 6. Line 1

Compression-shear coupling effort ?

Page 6 line 4 from bottom

Meaneffectivestress. Please revise mean effective stress

Table 1

It is available to assume any parameters in modelling.  Other hands, source reason is important to accept from reader. Then, K and G were constant, respectively. Please mention source reason or add accurate references.

Table 1

Evaluation of liquefaction or past liquefaction deformation for sand generally require obtaining relative density value according to each void ratio.

Figure 1 to 3

Modified TTS had advantage to critical stats that was understood from stress-strain curves in Figs 1 to 3. However, reviewer found out some afraid points, which large difference between measurement results and simulation data sets close to small strain in stress-strain behavior such as elastic strain raging.

Please explain to be cause.

Figure 4

Relationship between void ratio and mean effective stress. These data sets could be draw in straight line or smoothly line with curve.

Also, please make relationship between relative density and mean effective stress as modified ideas.

Page 8

Line 2 from top

Authors mentioned below;

and the shear stress in the simulation results is consistent with the experimental result.

However, due to considering results Figs 1 to 3, this sentence is NOT accurate.

3.3 Loading process loading

Comparison between simulation and measurements were indicated at stress ranges from 3000 kPa and 2000 kPa. Authors distinct “low stress level”.  Why this experimental work choice further high stress level such as 3000 kPa.

Conclusion

Loading/unloading criteria as one of constitutive model factors was focused.  However, loading/unloading process were NOT explained in experimental works.  Also, did authors introduce and explained effect of loading/unloading process in stress history.

Conclusion

Term “friction angle in critical state” was referred in conclusions.  Reviewer could not find out evaluated number or measured value regard to friction angle from triaxial test. 

Reviewer 2 Report

This paper addresses two interesting topics in soil mechanics - namely the

prediction of critical state without an elasto-plastic framework and the

prediction of phase transitions in the low effective stress regime. The

investigation assumes incompressible solid and fluid phases, effectively

focusing on the changes in the skeleton of the sample itself. 

This paper approaches this complex topic from the viewpoint of non-equilibrium

thermodynamics. Specifically, it implements a simple version of the Tsinghua 

Thermodynamics Soil Model (TTS) and establishes the need for a mitigation coefficient

to counter the effect of elastic relaxation. The paper asserts that a coupling 

energy mechanism is essential in granular materials. Further elaboration would

improve readability.

If the reader is not familiar with the TTS model, the mention of non-equilibrium 

thermodynamics may be disruptive. Non-equilibrium thermodynamics is a broad and 

evolving field and the paper uses this term loosely. A specific definition of 

which aspect of this field is relevant would improve the paper. Referring to 

near equilibrium and conventional state variable methodology would also improve 

the paper.

The paper discards the thermodynamic models in the literature without identifying 

the authors' rationale for ignoring them. A paragraph comparing the dis/advantages

of the two approaches in a discussion section before the conclusions would speak to

this oversight.

The introduction uses the term coupling loosely. There are many forms of coupling in

thermodynamics. The precise meaning only becomes clear once the text refers to the

viscosity relation. Adding an adjective in the introduction would improve readability.

The introduction discards the elastic model, but eq 14 is a linear elastic model.

Eq 14 does not account for variations in density as is necessary in soil mechanics.

Some comment on both these points would help.

Although the paper mentions interlocking, which covers the intermediate range 

between zero effective pressure and critical state pressure, the paper does

not elaborate on this interlocking. It could be improved with a reference to 

Taylor (cited in ref 19) and Schofield (2006) and a section discussing this

topic. The paper mentions the increase in friction angle, which is relevant

here.

The phase transition point is not clearly defined if compared to the conventional

understanding of critical state. Is a critical state a phase transition point?

A mental model or physical description of the coupling mitigation coefficient for

the uninitiated would help.

Several references listed in the paper are not cited in the body. If not cited,

they should not be included in the body.

With more work along these lines, this paper would reach a broader audience.

Author Response

Round  2

Reviewer 1 Report

Revised sentences are satisfied.

Reviewer 2 Report

Your changes have made your paper more transparent technically to an outside reader. Some minor editing by an English expert would improve the flow of your paper. You may wish to pass this draft to someone within your institution.